

# AerGOM, an improved algorithm for stratospheric aerosol retrieval from GOMOS observations. Part 2: Intercomparisons

Charles Étienne Robert[1], Christine Bingen[1], Filip Vanhellemont[1], Nina Mateshvili[1],
Emmanuel Dekemper[1], Cédric Tétard[1], Didier Fussen[1], Adam Bourassa[2], and Claus Zehner[3]

[1]Institut d'Aéronomie Spatiale de Belgique, Brussels, Belgium
[2]Institute of Space and Atmospheric Studies, University of Saskatchewan, Saskatoon, Canada
[3]ESA European Space Research Institute, Frascati, Italy

*Correspondence to:* C. E. Robert (charles.robert@aeronomie.be)

**Abstract.**

AerGOM is a retrieval algorithm developed for the GOMOS instrument onboard Envisat as an alternative to the operational retrieval (IPF). AerGOM enhances the quality of the stratospheric aerosol extinction retrieval due to the extension of the spectral range used, refines the aerosol spectral parameterisation, the simultaneous inversion of all atmospheric species as well as an improvement of the Rayleigh scattering correction. The retrieval algorithm allows for a good characterisation of the stratospheric aerosol extinction for a wide range of wavelengths.

In this work, we present the results of stratospheric aerosol extinction comparisons between AerGOM and various space-borne instruments (SAGE II, SAGE III, POAM III, ACE-MAESTRO and OSIRIS) for different wavelengths. Due to the unique observational technique of GOMOS, some of the results appear to be dependent on the star occultation parameters such as star apparent temperature and magnitude, solar zenith angle, latitude and obliquity. A systematic analysis is carried out to identify biases in the dataset, using the various spaceborne instruments as references. This bias characterization is extremely important for data users and might prove valuable for the production of unbiased long-term merged dataset.

## 1 Introduction

Stratospheric aerosols are an important part of the Earth system due to their impact on the planet's radiative balance and the crucial role they play in heterogeneous chemistry (Solomon et al., 2011). They can be produced either via binary homogeneous nucleation of $H_2SO_4$ and $H_2O$ close to the tropical tropopause (so-called background aerosols) or during volcanic eruptions, and form the so-called Junge layer, extending from the tropopause to approximately 35 km (Junge et al., 1961).

In order to better understand their behaviour and evolution, it is critical to observe these particles globally and over an extended period of time. Various techniques have been employed to retrieve stratospheric aerosols such as solar occultation (e.g., Kent and McCormick, 1984; Thomason et al., 2008; Randall et al., 2001; Thomason et al., 2010; Sioris et al., 2010), balloon-borne measurements (Hofmann et al., 1975; Renard et al., 2002; Deshler et al., 2003), satellite limb sounding (Taha et al., 2011; Bourassa et al., 2007), ground-based lidar (DeFoor et al., 2012; Jäger, 2005), and twilight brightness variation (Mateshvili, 2005).



Another measurement technique, stellar occultation from space, was utilized by the Global Ozone Monitoring by Occultation of Stars (GOMOS). This instrument collected transmission spectra from the Earth's limb in the UV-Vis-NIR, allowing the retrieval of atmospheric profiles from various species, such as $O_3$, $NO_2$, $NO_3$, as well as aerosol extinction profiles (Kyrölä et al., 2004; Bertaux et al., 2010). These species are currently retrieved by the latest GOMOS operational data processing algorithm (IPF v6.01).

Recently, a new stratospheric aerosol retrieval algorithm called AerGOM, extensively covered in a companion paper (Vanhellemont et al.), has been applied to the GOMOS transmission data in order to obtain improved stratospheric aerosol profiles. AerGOM is currently the main algorithm used to produce the stratospheric aerosol dataset for the Aerosol Climate Change Initiative (Aerosol_CCI), an ESA project focusing on both tropospheric and stratospheric aerosols (Holzer-Popp et al., 2013).

The purpose of this paper is to assess the agreement and discrepancy between AerGOM stratospheric extinction measurements at different wavelengths and those of various spaceborne instruments that observed the stratosphere in the same spectral range during the Envisat mission, namely SAGE II, SAGE III, POAM III, MAESTRO and OSIRIS. Beside the general comparison between AerGOM and other instruments, the influence of various stellar occultation parameters such as obliquity, star magnitude and temperature, solar zenith angle as well as the spatio-temporal variability is studied.

## 2   The GOMOS Instrument

GOMOS was on-board the successful ESA Environmental Satellite (Envisat) mission. Envisat payloads gathered information about the state of the Earth's atmosphere from shortly after its launch in March 2002 until communication was lost with the satellite in April 2012. The GOMOS instrument functioned almost continuously during its lifetime, except in 2005 when problems with the instrument forced the ground segment team to switch to the redundant measurement system due to errors with the scanning mirrors, impacting measurements during several months (ESA, 2007).

The instrument measured the light transmission from up to 300 stars through the Earth's atmosphere using 4 spectrometers covering the following spectral regions: 248-371 nm, 387-693 nm, 750–776 nm and 915–956 nm. The vertical sampling ranges from 200 m to 1.7 km, depending on the obliquity and the tangent altitude of the observation.

The starlight transmission is not only affected by scattering and absorption, but also modified by refractive effects such as chromatic refraction and refractive dilution. More problematic for the analysis of the transmission spectra however is scintillation, i.e. random fluctuations in the measured intensity of stellar light caused by refractive irregularities due to atmospheric instability. Two fast photometers measuring in the blue (473-527 nm) and the red (646-698 nm) part of the visible spectrum were used for the scintillation correction and also provided high-resolution temperature profiles (Sofieva et al., 2009).

Beyond these issues, the uncertainty of the retrieval is largely determined by the temperature and magnitude of the observed star. Even bright stars are point sources of low-intensity compared with the sun. Hence, profiles obtained from stellar occultations have larger uncertainties compared with solar occultation measurements. However, this drawback is compensated by the fact that stars are abundant in the sky: 30-40 occultations/orbit have been typically performed (compared with the 2 occultations available in the case of solar occultation), although this number decreased to 20-30 occultations/orbit after the instrument





malfunction in 2005. The retrieval of species using stellar occultation is possible in both bright and dark limb, but in the case of bright limb geometry, the weakness of the signal compared to the ambient light makes the retrieval even more challenging. At this stage, bright limb measurements are not used for the retrieval of stratospheric aerosol extinctions.

### 2.1 GOMOS operational stratospheric aerosol retrieval

5 The GOMOS operational stratospheric aerosol extinction is retrieved in a two-step process as described in Kyrölä et al. (2010). The first step consists of the spectral inversion, where measured transmittance spectra at each tangent altitude are inverted to slant path integrated column densities/optical thicknesses (for gases and aerosols, respectively). The second step is the spatial inversion, where the slant path columns for each species are inverted to local concentration/extinction profiles. The spatial inversion uses the Tikhonov altitude smoothing technique (Twomey, 1996; Rodgers, 2000) to remove the residual scintillation 10 perturbations in measurements.

The current choice of the Tikhonov parameters leads to the removal of all strong oscillations, at the cost of the vertical resolution, chosen as 4 km (Vanhellemont et al., 2010).

The specification of the aerosol scattering cross-section is difficult since the aerosol content may be very different depending on the state of the atmosphere and the nature of the dominant aerosol mode (background, volcanic, etc). The current (v6.01) 15 spectral inversion assumes that the stratospheric aerosol extinction obeys a quadratic polynomial as a function of wavelength:

$$\beta(\lambda) = \sigma_{\mathrm{ref}}(c_0 + c_1(\lambda - \lambda_{\mathrm{ref}}) + c_2(\lambda - \lambda_{\mathrm{ref}})^2) \tag{1}$$

where $c_0$, $c_1$ and $c_2$ are coefficients to be retrieved, and $\lambda_{\mathrm{ref}}$ is a reference wavelength arbitrarily fixed at 500 nm. This is a versatile approach that can represent large and small particle spectra within good approximation. Rayleigh scattering is not retrieved directly from the measurements, but removed using external European Centre for Medium-range Weather Forecasts 20 (ECMWF) air density data. This approach sidesteps problems of interferences with the residual scintillation and the spectrally similar aerosol contribution. $NO_2$ and $NO_3$ are retrieved separately using a DOAS approach (Hauchecorne, 2005).

The resulting stratospheric aerosol extinction profiles are of good quality around 500 nm, despite being oversmoothed. At other wavelenghts, the profile quality is poor. The main reason for this is that only the coefficient $c_0$ is directly smoothed by the Tikhonov approach. This makes the extinction very noisy when departing from the reference wavelength.

25 Aerosol extinction relative error estimates for bright stars (providing the best signal-to-noise ratio) are of the order of of 30% at 10 km, 2-10% from 15 to 25 km, and 10-50% from 25 to 40 km (Vanhellemont et al., 2010). The extinction profiles become increasingly uncertain at lower tangent altitudes because the transmitted light becomes weaker due to increasing atmospheric absorption by gases, aerosols and clouds.

### 2.2 AerGOM Stratospheric Aerosol Retrieval

30 AerGOM is an improved stratospheric aerosol extinction retrieval method developed for the GOMOS experiment and designed to rectify some of the problems of the operational retrieval, namely the difficulty to obtain proper stratospheric aerosol extinction profiles at $\lambda \neq 500$ nm and the inadequate error characterization of the extinction.





The most important improvements implemented are: 1) the extension of the spectral range used for the retrieval using spectrometer B1 (750–776 nm); 2) the refinement of the aerosol spectral parameterization using a $2^{nd}$ order polynomial in $\lambda^{-1}$; 3) the simultaneous retrieval of all species ($O_3$, $NO_2$, $NO_3$, aerosols); 4) a better Rayleigh scattering correction by considering the spectral dependence of the King factor $F_{air}$ (Bodhaine et al., 1999); 5) the inclusion of covariances between species after

spectral inversion. A detailed description of the algorithm and its improvements is given in a companion paper (Vanhellemont et al.).

The main steps of the AerGOM algorithm are similar to the operational retrieval. First, GOMOS transmittance data are read, along with the ECMWF temperature and pressure profile coincident with the stellar occultation measurements. Based on this data, temperature-dependent gas absorption cross-sections are calculated for each tangent height to create the spectral

matrix. One can choose either to calculate the Rayleigh scattering contribution based on the ECMWF data or to retrieve it along with the other species. Climatological profiles of various species are provided as a starting point for the non-linear Levenberg-Marquardt spectral inversion at the highest altitude. For all levels below, the retrieval from the previous altitude is used as a priori, leading to slant path integrated column densities and aerosol optical thicknesses at each tangent height. This is finally followed by a spatial inversion using the Tikhonov approach that leads to local aerosol extinctions, along with

the density profiles of the different gaseous species considered. It should be noted that the Tikhonov parameters used for the spatial inversion can be tuned to optimize the removal of residual scintillation. In particular, imposing weak regularization to gaseous species with respect to particulate species leads to noisier profiles for gas concentrations but smoother and more realistic profiles for aerosols.

The improved aerosol spectral law in AerGOM is more flexible since the polynomial can be of any degree and can be based

on either $\lambda$ or $\lambda^{-1}$. The formulation as a spectral interpolation formula between a number of discrete extinction coefficients $\beta(\lambda_i)$ that are to be retrieved is also better conditioned and physically more clear than for the operational retrieval.

For the quadratic spectral law, this gives:

$$\beta_{aero}(\lambda, r) = \sum_{i=1}^{3} q_i(\lambda) \beta_{aero}(\lambda_i, r) \tag{2}$$

with

$$q_i(\lambda) = \frac{(\lambda - \lambda_j)(\lambda - \lambda_k)}{(\lambda_i - \lambda_j)(\lambda_i - \lambda_k)} \tag{3}$$

with $\lambda_i$, $\lambda_j$, $\lambda_k$, different wavelengths to be specified ahead of time.

For more details on the AerGOM retrieval algorithm, we refer the interested reader to section 3 of the accompanying paper (Vanhellemont et al.).





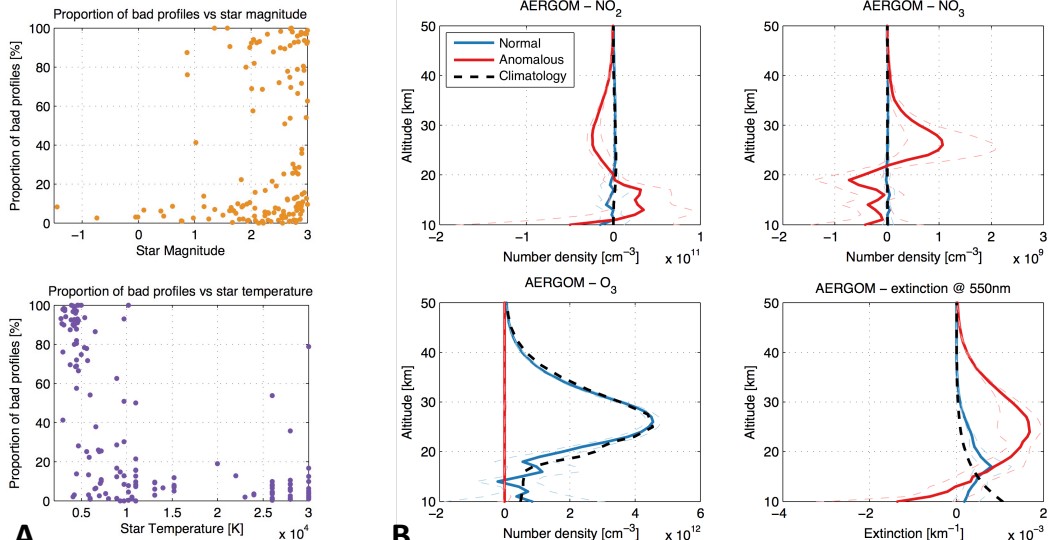

**Figure 1.** A) Proportion of bad profiles as a function of star temperature and magnitude. B) Median gas and aerosol extinction profiles for normal and anomalous AerGOM retrievals.

### 2.3 Anomalous Profiles and stellar occultation parameters

During the development phase of AerGOM, it was discovered that while the algorithm has beneficial properties regarding the retrieval of stratospheric aerosol profiles, it does have a drawback compared to the operational algorithm, namely that some of the converging retrievals exhibit some non-physical behaviour leading to incorrect aerosol extinction profiles. These 5 so-called "anomalous profiles" are mostly retrieved for occultations carried out with either a dim ($M_{star} > 2$) and/or a cold ($T_{star} < 5 \times 10^3$ K) star, as shown in Figure 1a.

The reason for the retrieval of such profiles is probably due to low signal-to-noise ratio (SNR) of the transmittance at shorter wavelengths for such stars, leading to an incorrect retrieval of $O_3$, $NO_2$ and $NO_3$ as shown in Figure 1b. These profiles are easy to identify and were discarded for the intercomparisons of this paper, but this finding prompted the consideration that 10 some of the retrievals might be affected by occultation parameters such as star properties, solar zenith angle (SZA) that could lead to straylight, and occultation obliquity which is an important factor in the imperfect correction of atmospheric scintillation (Sofieva et al., 2009). Therefore, another aspect of this inter-comparison involves studying the consistency of the agreement of AerGOM aerosol retrievals with those of other instruments under various occultation conditions. Section 6 presents the results of these comparisons.





**Table 1.** Characteristics of the stratospheric aerosol extinction datasets used in this work.

| Instrument | Version | Host satellite | Measurement method | Time coverage | Aerosol extinction wavelength(s) available [nm] |
|---|---|---|---|---|---|
| SAGE II | 7.0 | ERBS | solar occultation | 1984/10/24 — 2005/08/22 | 386, 452, 525, 1020 |
| SAGE III | 4.0 | METEOR 3M | solar occultation | 2002/02/27 — 2004/12/02 | 384, 448, 520, 601, 675, 755, 869, 1022, 1545 |
| POAM III | 4 | SPOT-4 | solar occultation | 1998/04/22 — 2005/04/04 | 354, 439, 602, 779, 922, 1020 |
| MAESTRO | 3.12.1 | Scisat | solar occultation | 2004/02/21 — now | 525, 530, 560, 603, 675, 779, 875, 922, 995, 1012 |
| OSIRIS | 6 | Odin | limb | 2001/10/28 — now | 750 with Ångström coefficient |
| GOMOS (IPF) | 6.01 | Envisat | stellar occultation | 2002/04/15 — 2012/04/08 | 350 - 750 |
| GOMOS (AerGOM) | 1.0 | Envisat | stellar occultation | 2002/04/15 — 2012/04/08 | 350 - 750 |

## 3 Inter-comparisons instruments

It has been pointed out by Thomason et al. (2010) that aerosol validation is challenging because there is no standard measurement with which to compare. Occultation instruments often validated aerosol data by comparing with each other and a small number of other space-based instruments. A difficulty encountered with this approach is that the aerosol extinction measurements in one experiment do not always have their spectral counterpart in other experiments and cannot be directly compared. Another possibility would be to perform a validation with lidar measurements. Although it could prove useful for periods following volcanic eruptions, it would be non-trivial for periods of low stratospheric aerosol loading, as the corresponding lidar backscatter ratios are too small to convert them to extinction with the required precision/accuracy for validation.

In this paper, we opted for the approach of comparing measurements with multiple instruments datasets. The power of this approach is that by uncovering similar and/or consistent features across the many available measurements, some sort of consensus can be reached on the agreement of the data. The weakness of such methodology is that there is no clear independent source of high quality information, and consensus does not imply any form of absolute truth.

For this work, the SAGE II, SAGE III, POAM III, MAESTRO and OSIRIS instruments are used as a basis for the inter-comparison efforts with AerGOM. Table 1 provides some general information about these instruments and their respective stratospheric aerosol products.

Figure 2 shows the spatio-temporal coverage of the datasets used for this study. Note that the color scale indicating zonally averaged observations per month in a 10° latitude bin is different for each experiment. There is a vast difference (a factor of 3-4) in coverage from a limb instrument (OSIRIS) compared with solar occultations experiments. GOMOS coverage is more extensive than what is shown in Figure 2 for AerGOM, but to ensure high-quality data, all observations that could potentially be straylight-contaminated were filtered out resulting in a limited coverage at high latitudes. It is also important to note that some solar occultation instruments have a limited latitude coverage, (e.g. SAGE III, POAM III), which can sometimes limit the generalization of the results from the comparisons since they might be spatially biased.





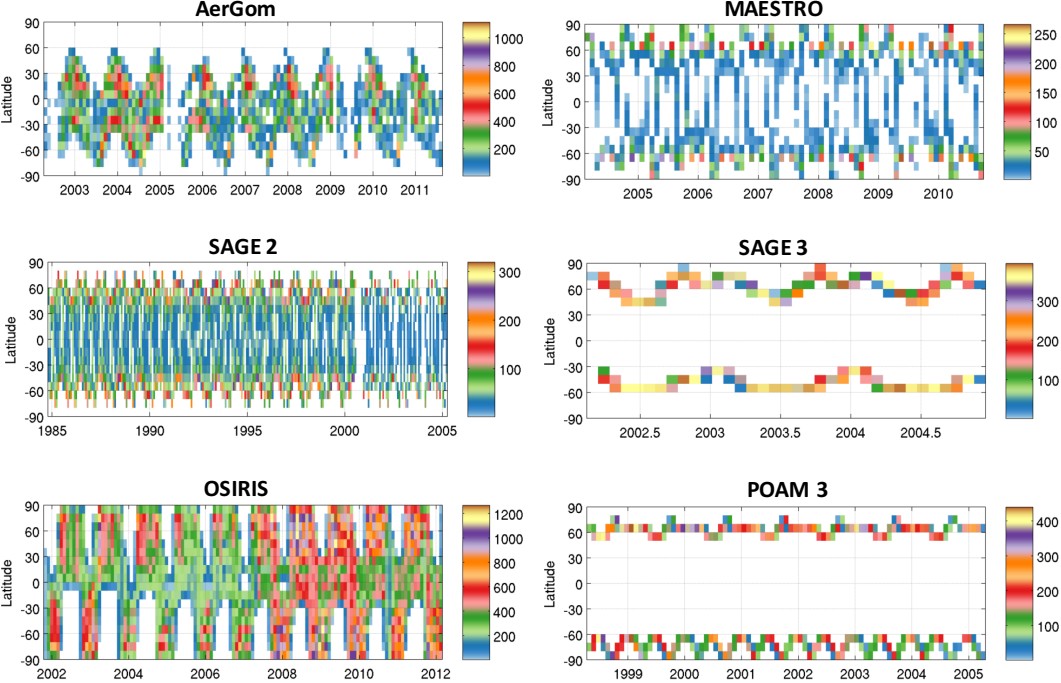

**Figure 2.** Latitude and temporal coverage for the various instruments used for the comparisons. The number of observations per month is calculated for a $10°$ latitude bin. The color code gives the number of observations per month.

## 3.1 SAGE II

The Stratospheric Aerosol and Gas Experiment (SAGE II) on-board the Earth Radiation Budget Satellite (ERBS) provided high-quality vertical profiles of important atmospheric species from the mid-troposphere through the stratosphere during a mission that lasted from October 1984 until August 2005. The instrument recorded the attenuation of sunlight by the Earth's

5   atmosphere in seven spectral channels between 386 nm and 1020 nm during each sunrise and sunset encountered by the spacecraft. The measurements were separated into slant path optical depth contributions for $O_3$, $NO_2$, $H_2O$ and aerosol at four channels (386, 452, 525 and 1020 nm) using a least-squares technique (Thomason et al., 2008).

In this work, we use the SAGE II v7.0 stratospheric aerosol dataset (Damadeo et al., 2013) for which the 386 nm aerosol channel is not recommended due to some unexplained contribution that can be substantial (approaching 30%) at low extinction

10   levels. Therefore, this channel is not considered in the comparisons. Note also that the aerosol extinction coefficient measurements at 452 nm do not reliably extend below 12 km and will not be used below this height, whereas measurements made at 525 nm are reliable in the UTLS and available as low as 5 km despite substantial impacts by ozone absorption and molecular scattering (Thomason and Vernier, 2013).



## 3.2 SAGE III

SAGE III was launched in December 2001 on-board the Russian METEOR 3M spacecraft. It gathered data from February 2002 until the end of the mission in March 2006, using the technique of solar occultation. It observed the line-of-sight (LOS) transmission profiles from 0.5 to 100 km at 87 wavelengths from the ultraviolet to the near infrared with an estimated 0.7 km vertical resolution.

Aerosol extinction is derived in nine spectral channels by removing the effects of molecular scattering, $O_3$ and $NO_2$ absorption. The precision and accuracy of the aerosol product is linked to the measurement noise in the channel, the quality of the Global Modeling and Assimilation Office (GMAO) density product, the noise and bias in the retrieved $O_3$ and $NO_2$, and the consistency of the cross-sections used in the $O_3$ / $NO_2$ multi-linear regression retrieval and those at the aerosol channel wavelengths.

It was found that the aerosol extinction coefficient measurements at 448, 520, 755, 869, and 1021 nm are reliable with accuracies and precisions on the order of 10% in the 15-25 km range (Thomason et al., 2010) . It is recommended to only exploit the 385 nm measurements above 16 km where the accuracy is on a par with other aerosol channels. The 601 nm measurement is much noisier than other channels (approx. 20%) and it is suggested to use with caution. The 676 nm data is clearly defective, particularly above 20 km (accuracy as poor as 50%) and the precision is also low (approx. 30%). Therefore, these two channels will not be considered in the present study.

## 3.3 POAM III

The Polar Ozone and Aerosol Measurement Instrument (POAM III) (Lumpe et al., 2002; Randall et al., 2001) was launched in March 1998 on the Satellite Pour l'Observation de la Terre (SPOT 4) in a sun-synchronous polar orbit.

The instrument used the solar occultation technique to measure atmospheric transmission across nine spectral channels in the UV-Vis range. From these measurements, $O_3$, $NO_2$, $H_2O$ and $O_2$ vertical profiles can be retrieved. Stratospheric aerosols are also retrieved at several wavelengths (354, 439, 602, 778, 922, 1020 nm) up to an altitude of approx. 25 km.

POAM III sunrise aerosol extinction measurements at both 1020 nm and 450 nm are within $\pm$ 30% of SAGE II. However, POAM III exhibits a significant sunrise/sunset bias in its extinction measurements that leads to poorer agreement between SAGE II and the POAM III sunset data. This is important for this work since collocations with GOMOS observations are only found for POAM III sunset occultations. POAM III sunset aerosol extinction at 1020 nm and 440 nm both exhibit a positive bias with respect to SAGE II, which magnitude changes with altitude but can be as large as 50%.

## 3.4 MAESTRO

The Atmospheric Chemistry Experiment (ACE) mission (Bernath et al., 2005) was launched on 12 August 2003 on-board the SCISAT satellite and is still currently operational. The satellite is in a low-Earth circular orbit at an altitude of 650 km and 74° inclination. The ACE mission is comprised of two instruments: a Fourier transform spectrometer (ACE-FTS) and the Measurement of Aerosol Extinction in the Stratosphere and Troposphere Retrieved by Occultation instrument (MAESTRO) (McElroy





et al., 2013, 2007). The MAESTRO instrument uses the solar occultation technique and is made of two independent spectrophotometers, one measuring in the UV (285-550 nm, 1.5 nm spectral resolution) while the other observes in the VIS-NIR spectral region (525-1020 nm, 2 nm spectral resolution). These measurements allow the retrieval of atmospheric species such as $O_3$, $NO_2$, $H_2O$, $O_2$ and aerosols. Measurements are made at tangent altitudes between 0 and 150 km (using measurements

between 100 and 150 km to determine the sun reference spectrum) and allow for a best-case vertical resolution of 1.2 km at a tangent height of 22 km. The aerosol extinction is retrieved at 525, 530, 560, 603, 675, 779, 875, 922, 995 and 1012 nm wavelengths. Cirrus clouds are not filtered from the dataset.

Though able to retrieve high-resolution aerosols extinction profiles, MAESTRO has two issues affecting its measurements: 1) an altitude assignment problems that can lead to outlier data and 2) an unidentified problem (suspected to be a dark count

model issue) that makes small optical depths too large.

## 3.5 OSIRIS

The Optical Spectrograph and InfraRed Imaging System (Llewellyn et al., 2004), on-board Odin, measures the vertical distribution of atmospheric limb radiance spectra. The satellite was launched in February 2001 in a sun-synchronous polar orbit and continues full operation at the time of this writing. The local time of the ascending node is 18:00 LT, providing measurements

of the sunlit summer hemisphere, global measurements during equinox, and a limited coverage of the winter hemisphere.

The two sub-systems of OSIRIS are an optical spectrograph (OS) and an infrared imager (IRI). The optical spectrograph consists essentially of a grating and a CCD detector, and measures the limb radiance spectra from 280 to 800 nm with a spectral resolution of approximately 1 nm (Bourassa et al., 2007). The sampling resolution of the measurements is approximately 2 km. The IRI is composed of three vertical near-infrared channels that capture one dimensional images of the limb radiance at

1.26, 1.27, and 1.53 $\mu$m at a tangent altitude resolution of approximately 1 km. The previous version (v5) of the dataset was derived using only the optical spectrograph measurements, but now both instruments are used for the latest retrieval (v6) of the aerosol extinction profiles (Rieger et al., 2014), allowing a better characterization of the aerosol scattering phase function and improving the retrieval substantially. The drawback of this new approach is that retrievals are noisier and have a tendency to saturate at low altitudes and high aerosol loadings such as in the centre of volcanic plumes. It is this latest version (v6) of the

OSIRIS dataset that is used in this study.

It is important to note that the OSIRIS retrieval is based on measured radiances at 750 and 1530 nm, from which an Ångström exponent is derived. In this work, we perform comparisons with OSIRIS extinction at 750 nm, but also at wavelengths outside the range used to determine the Ångström exponent (350 and 550 nm).

## 4 Methodology

The comparison of datasets is based on the statistical analysis of collocated events, defined here as observations within a distance $\Delta r = 500$ km and a period $\pm \Delta t = 12$ h from each other. Since stratospheric aerosols are assumed to be slowly varying over time and space in the absence of volcanic activity, these criteria are deemed acceptable. It should be noted that processes





such as pyro-convective events, other tropospheric intrusions and polar stratospheric clouds can also change significantly the extinction signal in the stratosphere and therefore make it more difficult to compare data points that are further apart. The key is to strike a balance between the proximity of observations in time and space and the number of sample observations available for analysis. Evaluation of different collocation criteria showed that constraining further the $\Delta t$ and $\Delta r$ generally does not

affect the final results, but sometimes lead to undersampling.

The relative difference between the AerGOM extinction $\beta_{\mathrm{AerGOM}}$ and the extinction from a collocated measurement $\beta_i$ from dataset $i$ (in %) is:

$$100 \times \left( \frac{\beta_{\mathrm{AerGOM}} - \beta_i(\lambda)}{\beta_i(\lambda)} \right) \tag{4}$$

All observations with a relative uncertainty larger than 100% are discarded before performing the analysis in order to avoid

biasing the results due to inferior quality data, but it should be noted that using all observations does not alter the results significantly. The profiles are interpolated on a common 1 km spacing vertical grid using a linear interpolation method. AerGOM extinctions are interpolated at the wavelength(s) of the other instruments using Eq. 2. In this way, a distribution of values is obtained for each tangent altitude $z$ and wavelength $\lambda$ and the final results are derived from this distribution by calculating the interquartile mean and the semi-interquartile range, which should be robust estimates of the average value and the variability,

respectively.

## 5   Comparison of Collocated Profiles

Figure 3 shows the results of the intercomparison of AerGOM against all datasets from Table 1 using the method outlined in Section 4. Three different aspects of the comparison are shown: the relative difference (interquartile mean), the relative difference variability (semi-interquartile range), and the absolute aerosol extinction profiles of AerGOM and the other datasets

(interquartile mean). The total number of collocation is also indicated and varies widely from one dataset to the next. To quantify the effect of the change in retrieval algorithm from IPF to AerGOM, the comparisons were also performed using IPF and the results are shown as dashed lines in the relative difference and the variability plots.

Overall, the agreement between AerGOM and other datasets for tangent altitudes between 15 and 30 km is typically within $\pm 50\%$ for extinctions in the 400-600 nm spectral range. The comparisons are especially favorable with SAGE II and OSIRIS

between 20 and 30 km.

However, AerGOM aerosol extinction profiles for $\lambda > 700$ nm present a strong negative bias above 25-30 km with respect to all other datasets, increasing towards higher altitudes. The AerGOM absolute extinction profiles at $\lambda > 700$ nm also shows more structure than that of the other datasets, with a small peak around 16 km, and throughs near 13 and 20 km.

Comparisons between AerGOM extinction at shorter ($\lambda < 700$ nm) wavelengths and SAGE II, SAGE III and POAM III

extinction shows a strong positive bias below 20 km, increasing with decreasing altitude. This positive bias is larger for shorter wavelengths. These features could be the result of subvisible cirrus clouds present in the field of view (FOV), but it is unclear why only these datasets are affected while MAESTRO and OSIRIS show no such large positive bias and also why



the effect is much more pronounced in the case of AerGOM than for comparisons with IPF. The latter results suggest that this effect is mostly due to AerGOM retrieval algorithm and not the GOMOS instrument itself, despite its known decreasing SNR with decreasing tangent altitude. There was no cloud filtering performed for other datasets that could explain these large discrepancies. It is also not clear why the overall effect of clouds in the FOV would lead to a larger extinction for AerGOM

when compared to other datasets, except to presume that AerGOM has a greater sensitivity to clouds.

Another interesting aspect of the results is that the variability of the AerGOM comparisons with other datasets is typically smaller than those made with IPF, especially between 15 and 30 km. This is, however, not the case for SAGE III above 25 km tangent altitudes and POAM III at tangent altitudes below 20 km, where interestingly enough these comparisons with AerGOM do not show a good agreement. For extinction comparisons in the 350-600 nm spectral range, the variability increases with

decreasing tangent altitudes, and is larger for shorter wavelengths. This is expected as it simply follows the spatial and spectral behaviour of the GOMOS SNR, and confirmed by the similar IPF comparisons variability. The variability of the comparisons at $\lambda > 700$ nm is less systematic but tends to increase dramatically with tangent altitudes above 20-25 km, correlated with the strong negative bias.

It should be pointed out that the results of the comparison between AerGOM and MAESTRO extinction profiles show

a different behaviour of the relative difference than seen in the other comparisons. This is especially the case for shorter wavelengths ($\lambda < 700$ nm) where a relatively constant negative bias of -35% to -50% can be seen extending between 10 and 25 km. Above 25 km, all AerGOM extinctions become increasingly negatively biased with regards to MAESTRO with increasing altitude and wavelength. Another surprising feature of the comparison is the small variability (25%) of the relative difference with AerGOM, almost constant between 10 and 25 km and for all wavelengths.

For reference purposes, we also show the comparison between AerGOM and IPF profiles using 20 000 randomly chosen profiles, spanning different geolocations and occultation parameters. Even though the raw data come from the same instrument, the difference in the comparisons can be substantial. These results show the systematic differences due to the algorithms, as the GOMOS data used is exactly the same.

## 6   Bias Variability with star and occultation parameters

Section 5 described AerGOM's bias relative to other instruments, but it did not take into account the very specific features of GOMOS which do not concern the other sensors but may dramatically affect the quality of AerGOM extinction. Specifically, the use of a wide range of stellar sources with very different characteristics, the subsequent low value of the SNR and the versatility of the occultation configuration reflected in the obliquity and the solar zenith angle may all affect the GOMOS measurements.

The purpose of this section is to perform a more detailed analysis and assess whether occultation parameters may affect the extent and consistency of the bias between AerGOM and other instruments. For each instrument, comparisons were carried out as explained in Section 4, except that only a specific subset of collocated profiles corresponding to particular criteria was used to calculate the interquartile mean. The parameters under investigation are: star properties, solar zenith angle, obliquity



**Table 2.** Classes of star properties (as defined in this work)

| Star Temperatures $10^3$ K | Descriptor | Star Magnitudes | Descriptor |
|---|---|---|---|
| 0 - 6 | cold | -1.5 - 1.5 | bright |
| 6 - 26 | mid-cold | 1.5 - 2.3 | mid-bright |
| 26 - 40 | hot | 2.3 - 3 | dim |

and latitude of observation. Note that this analysis is only valid for the AerGOM retrieval and cannot be generalized to the IPF dataset.

Studying the effects of a given occultation parameter or star property assumes that only one variable will change while all other parameters are constant, but this is not always the case for GOMOS. Most parameters are somewhat interdependent, albeit very loosely in some cases. Figure 4 gives an overview of the interdependence of several parameters: star magnitude, star temperature, latitude, SZA and obliquity. The figure shows 2D histograms of the number of observations for different combinations of these occultation parameters, taking into account only dark limb GOMOS observations. From these graphs, one can see for instance that at low SZA ($\leq 120°$), almost no bright stars are available and mostly only dim stars will be used for occultation. Maybe the clearest and most important dependence among the occultation parameters is observed for solar zenith angles and latitudes of observations, where low SZA values correspond to high-latitudes and equatorial observations to high SZA values. We must therefore use care when analyzing the results of comparisons with regards to certain occultation parameters and take into account this interdependence.

## 6.1 Star Properties

The properties of stars (star temperature $T_{\text{star}}$ and magnitude $M_{\text{star}}$) used as light source by GOMOS largely determine the shape of its spectral irradiance: cold (hot) stars have larger spectral irradiance at longer (shorter) wavelengths. In addition, the magnitude of the star might mitigate or aggravate the impact of the shape of the spectral irradiance on the quality of the retrieval by altering further the SNR in different spectral regions. In particular, it could be expected that dim stars seriously affect results at short (long) wavelengths for cold (hot) stars. Table 2 details the nine distinct categories of stars that we have defined for this work and that we will consider, ranging from dim and cold to hot and bright.

Figure 5 presents the results of the comparisons between AerGOM and the other datasets at various wavelengths and according to the defined star classes. The rightmost panel shows the number of observations available to perform the comparisons. Note that in some cases, the number of observations is very limited so that the effects seen in these cases may be strongly affected by subsampling.

There is a clear departure from the consensus across the various experiments at long wavelengths ($> 750$ nm) for dim hot stars occultations. More particularly, the effect is visible for occultations using stars with $M_{star} > 1.5$, except for cold stars. In these cases, AerGOM is more negatively biased than usual, starting around 25 km and worsening towards lower altitudes. The





star magnitude plays a major role on the bias variability for hot stars. Case in point, for the AerGOM-OSIRIS comparison at 750 nm, the bias can vary from -300% for dim/hot stars to +50% for bright/hot stars at 17 km.

The occultation star properties have the largest influence on AerGOM extinctions when considering dim/hot stars, especially for AerGOM aerosol extinction at wavelengths larger than 650 nm. The effect can also be seen at 550 nm, but to a more

limited extent. For SAGE II (452, 525 nm), SAGE III (520 nm), MAESTRO (525 nm) and OSIRIS (550 nm), AerGOM is more negatively biased between approximately 15-20 km.

At shorter wavelength (< 400 nm), the AerGOM comparison for cold stars occultations also shows a different behavior with respect to the other star properties comparisons, but the effect is much less dramatic and not consistent across the various datasets. For SAGE III, dim/cold stars occultations are very negatively biased between 17 and 23 km with regards to the other

occultations, but is positively biased between 25 and 35 km. Overall, in this short wavelength range, the weakness of the signal from dim/cold stars is responsible for the erratic behaviour of the bias profile at the lowest altitudes toward the troposphere.

## 6.2 Solar Zenith Angle

Another parameter studied is the solar zenith angle at which the stellar occultation was carried out. At low SZA, there is a larger probability that straylight finds its way into the instrument. SZA is also an indicator of the local time, although there is

no reason to believe that this can affect the extinction comparisons.

Figure 6 shows the results of the comparisons between AerGOM and other datasets acccording to different values of the SZA. The common feature to these comparisons is that AerGOM is more negatively biased for low SZA. The impact of the SZA on the bias seems to be progressive, what is clear when looking for instance at the AerGOM-OSIRIS comparisons for 750 nm. The SZA seems to impact the comparisons mostly above 15 km, although this could be linked to poor sampling below

that tangent altitude. The effect of the SZA on the comparisons at wavelengths around 500-600 nm seems to be minor, but gain in importance above 27 km.

## 6.3 Obliquity

The obliquity, combined with the wavelength and the altitude, is indicative of errors in the scintillation correction. For vertical occultations (obliquity = 0°), the correction should be accurate but the current scintillation correction is not able to remove the

wavelength-dependent distortion of transmission spectra caused by isotropic scintillations, which can be present in non-zero obliquity occultations. This distortion is a significant part of the error budget for bright stars (Sofieva et al., 2009), and it is interesting to see if this translates into systematic differences for the comparisons.

Comparisons between AerGOM and various datasets for different obliquities of occultation are presented in Figure 7. There are no single aspects which are common to all comparisons. For the SAGE II comparison, there seems to be a larger positive

bias in AerGOM observations for measurements with obliquity between -20° and 0°. It affects the comparisons at altitudes between 27 and 33 km, for both 452 and 525 nm. It seems unusual that it does not affect the data with an obliquity between 0° and 20° in the same fashion, as one would expect the effect to be symmetric, but it is clearly not the case. For large values of the obliquity ([80° - 90°]), AerGOM extinction is more negatively biased between 15 and 22 km, but the difference is modest.





For the remaining dataset comparisons, the effect seems to be mostly seen at large wavelengths, with a more negatively biased AerGOM for large values ([60° - 80°]) of the obliquity. This is mostly the case between 15 and 25 km. The amplitude of the bias varies with each dataset, with POAM III being most affected, followed by MAESTRO and OSIRIS.

## 6.4 Latitude

The latitude of the occultation can strongly affect comparisons, especially because cloud phenomena are involved. At high latitudes, polar stratospheric clouds (PSC) can affect the mean extinction profiles if not filtered out, while in the tropics and mid-latitudes, high altitude cirrus clouds can have an impact on the lower stratospheric extinction. Note that for the comparisons performed in this work, no filtering of cirrus or PSC was carried out. It should also be mentioned that there could be an indirect effect of the latitude on the extinction profiles due to the strong correlation between SZA and latitude (see Figure 4). The

influence of SZA on the extinction has already been examined in Section 6.2.

    Figure 8 shows that for SAGE II, MAESTRO and OSIRIS, there are some effects on the comparisons pertaining to the latitudes at which the occultations were made. The most prominent effect is seen in the tropics, for latitudes between -30° and 30°, where the comparisons show an increased bias between 15 and 20 km at all wavelengths. For SAGE II, the mid-latitude occultations also seem to be impacted, although to a lesser degree and at lower altitudes (13-17 km). Interestingly, von Savigny

et al. (2015) also reported similar bias in the lower stratosphere when comparing SCIAMACHY and SAGE II stratospheric aerosol extinction, although the bias increased with decreasing altitudes. This seems to indicate that cirrus clouds do have an impact on the comparisons. For OSIRIS, the aerosol data are not cloud filtered per se but they do provide a data set that is restricted to the stratosphere by cutting off profile information at the tropopause, which effectively work as a cloud filter in the tropics and might explain the positive bias between AerGOM and OSIRIS. For the other datasets, the reason for the fact that

AerGOM is more positively biased is unclear. There are also no indication that PSCs have an impact on the comparisons.

    The AerGOM-OSIRIS comparison at 750 nm also shows a notable difference for the southern hemisphere (latitude ≤ -30°) compared with the other latitude bands, with AerGOM data being more negatively biased in these cases case. This effect is also seen at shorter wavelengths between 12 and 20 km, but to a lesser extent. However, these differences could well be explained by the low SZA values of the observations in these categories (a majority of observations had a SZA ≤ 120°)

No results are shown for POAM III and SAGE III because the collocations with AerGOM were only contained within one latitude band.

## 7   Conclusion

In this study, we compared the results of stratospheric aerosol extinction coefficients retrieved by AerGOM in the UV-Vis with several datasets observing in the same spectral range and during the same period as GOMOS. Overall, features common to

intercomparisons with almost all sensors are:

    – an agreement within ±50% in the 400-603 nm spectral range, between 15 and 30 km.



- a strong positive bias below 20 km at $\lambda < 700$ nm , consistent with the weakening of the signal toward shorter wavelength

- strong negative bias above 700 nm at altitudes close to 30 km. The reason for this is not clearly identified, but could be due to the wrong attribution of extinction to aerosol and gases.

- consistent vertical variability of aerosol extinction above 700 nm across the instruments used for comparison.

Section 6 covered different aspects of the AerGOM data comparisons with other dataset as it related to the occultation parameters. More detailed analysis taking into account the star parameters (magnitude, temperature) and occultation characteristics (SZA, obliquity, latitude) leads to the following conclusions:

- The quality of the retrieval is mainly influenced by the star parameters that directly impact the SNR of the measurement. The dominant parameter is the magnitude quantifying the strenght of the star signal. Validation shows that a threshold of $M = 2.5$ is suitable for high quality retrievals. Hot stars perform better than cold stars and the recommended threshold is $T = 6 \times 10^3$ K.

- The second most important influence is the SZA. Using a threshold of $110°$ gives good quality results for extinction at $\lambda < 750$ nm but, at this stage, due to the bias problem observed above 25 km at longer wavelengths, one should use a threshold value of $130°$ for $\lambda \geq 750$ nm.

- The obliquity influences also the quality of the retrieval but to a lesser extent. Overall, a decrease of quality of the retrieval is only expected for a value of the obliquity above $60°$. Departing behaviour observed around $40°$ is most probably due to indirect effects of star properties, in view of the over-representation of cold dim stars at these values of the obliquity.

- The influence of the latitude of the collocation on the bias is mainly related to cloud detection that is not performed yet in AerGOM, in contrast to some of the other algorithms. The signature of cirrus clouds is clearly seen in the tropical region. No influence of PSC on the bias has been identified. Finally, the intercomparison between AerGOM and OSIRIS shows a particular behaviour in the 13-17 km altitude range, with an increasing negative bias polewards. This is probably due to the poor statistics in this case, and to the over-representation of very low values of the SZA and of dim stars in these regions.

These results can prove useful as guidelines to AerGOM data users as they shed light on the aspects of the occultations which might affect the results systematically.

*Acknowledgements.* This work was supported by a Marie Curie Career Integration Grants within the 7th European Community Framework Programme under grant agreement n°293560, the European Space Agency as part of the Aerosol_cci project and the Belgian Space Science Office (BELSPO) through the *Chercheur Supplementaire* programme. The AerGom project was financed by the European Space Agency (contract number 22022/OP/I-OL).





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




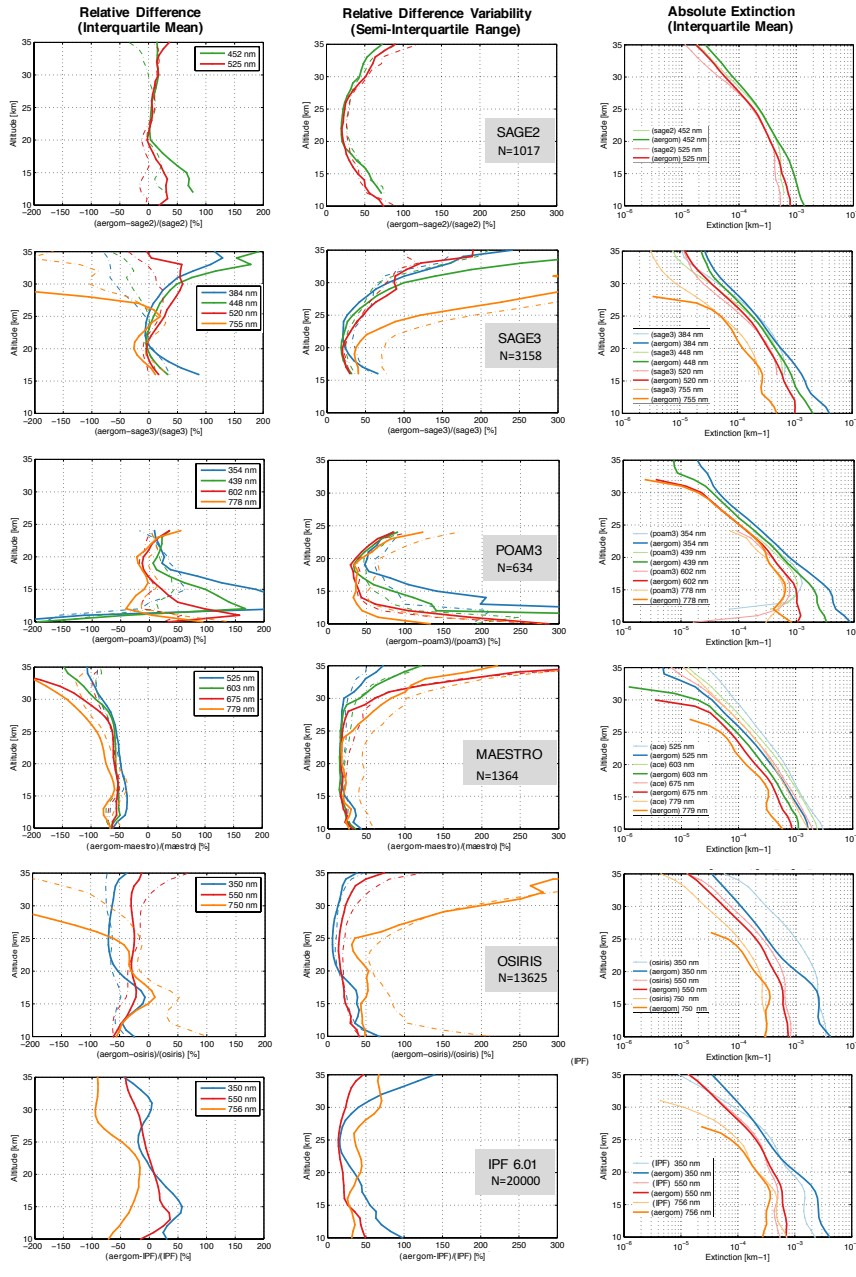

**Figure 3.** Relative difference interquartile mean (left panel), semi-interquartile range of the relative difference (central panel) and interquartile mean of the absolute aerosol extinction profiles (right panel) for each dataset at various wavelengths compared with collocated AerGOM profiles. The dashed curves in the relative difference plots were calculated using IPF instead of AerGOM. The total number of collocated profiles $N$ is also indicated.





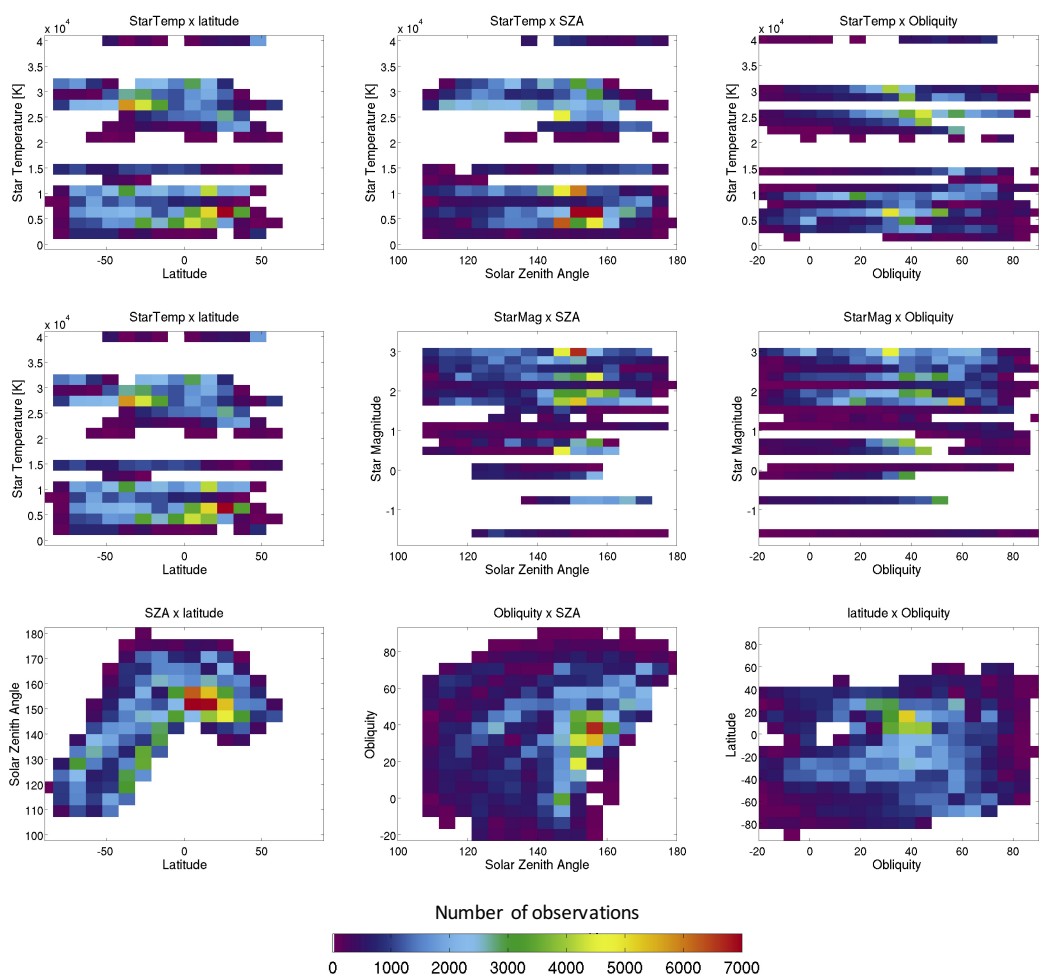

**Figure 4.** Interdependence of the GOMOS occultation parameters depicted using 2D histograms of the number of observations for different combinations of occultation parameters (star temperature and magnitude, solar zenith angle, obliquity and latitude of observation).





**Figure 5.** Relative differences between AerGOM and various datasets for varying star categories (left panel). The rightmost panel presents the number of available observations for each comparison and star category.





**Figure 6.** Relative differences between AerGOM and various datasets for different SZA of occultations (left panel). The rightmost panel presents the number of available observations for each comparison and SZA category.





**Figure 7.** Relative differences between AerGOM and various datasets for different obliquity of occultations (left panel). The rightmost panel presents the number of available observations for each comparison and obliquity category.





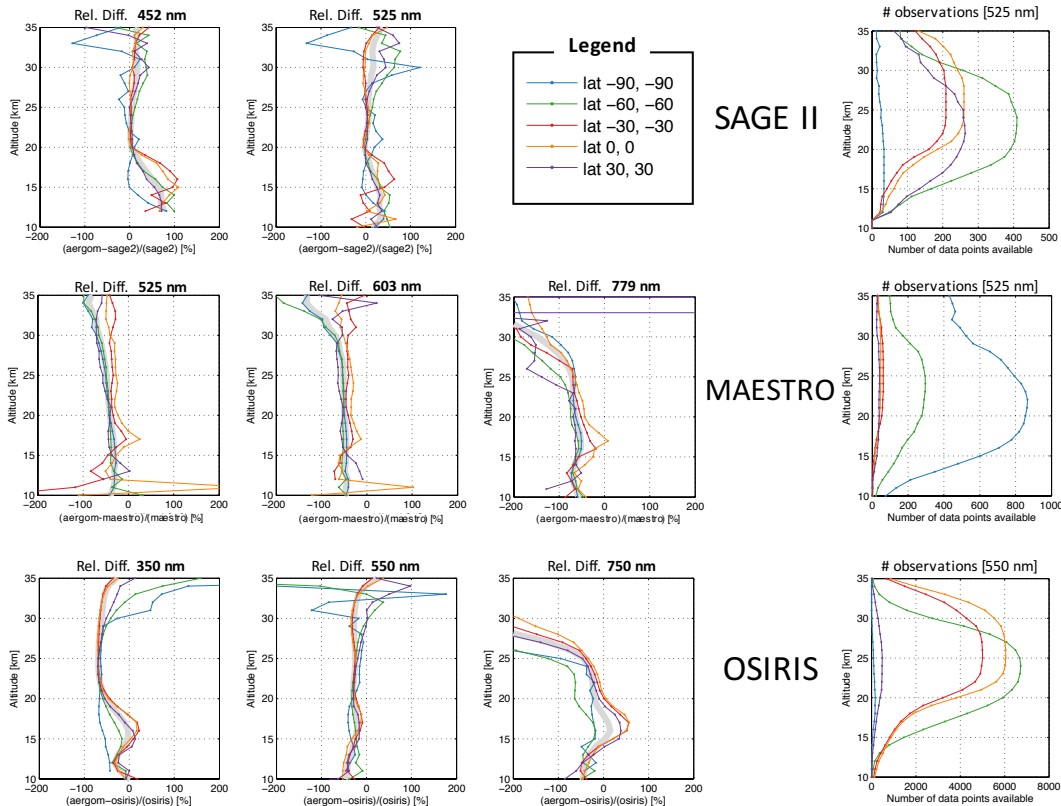

**Figure 8.** Relative differences between AerGOM and various datasets for different latitudes of occultations (left panel). The rightmost panel presents the number of available observations for each comparison and latitude category.