# Peer review of "AerGOM, an improved algorithm for stratospheric aerosol retrieval from GOMOS observations. Part 2: Intercomparisons"

_Atmospheric Measurement Techniques, 2016_

## Referee Comment (RC1) · S. Dörner (Referee) · 14 Apr 2016

This paper provides an overview on the performance of the new stratospheric aerosol extinction retrieval algorithm AerGOM by comparing the retrieval results with existing data sets from SAGE II, SAGE III, POAM III, ACE-MAESTRO and OSIRIS. A comprehensive comparison study is performed for different wavelengths at collocated measurement locations. A variety of parameters (latitude, sza, obliquity and star properties) indicate for which set of parameters the AerGOM retrieval has the best performance. The work is well structured and describes the comparison method in sufficient detail. The paper showed convincingly that AerGOM retrieval results are a valuable addition to the collection of stratospheric aerosol datasets.

I recommend for publication after two minor technical corrections as listed below.

- Page 2 Line 22: There are two different kinds of dashes here.

- Page 5 Figure 1: Depending on the PDF Viewer, the B side of the figure has a thin grey frame.

---

## Referee Comment (RC2) · Anonymous Referee #3 · 14 Apr 2016

The paper presents result of aerosol extinction coefficient comparison between the modified GOMOS retrieval (AerGOM) and various space-borne instruments at different wavelength. It also investigates the influence of various stellar occultation parameters. There is a separate paper that discusses the algorithm in more detail. The topic of the manuscript is of importance for the scientific community and suitable for publication in AMT. The paper should be accepted for publication after addressing the concerns mentioned below.

Major comments

The paper title clearly states that the analysis presented are for data Intercomparisons not validation. I suggest that the authors provide a proper validation and discussions of

the observed biases and data quality rather than just present data Intercomparisons. To do so, they need to use officially released and validated correlative measurements, and expand on the discussion, mainly in section 5. Furthermore, its not clear to the user what is the AerGOM end product, is it aerosol extinction at selected wavelengths or fitting parameters or both.

Abstract: Abstract should include summary of key findings.

Section 2: The authors need to describe the new data product format and retrieved aerosol wavelengths, if any.

Section 2.3, p5: "These profiles are easy to identify and were discarded for the inter-comparisons of this paper", is it also valid for section 6?

Section 3: Did you screen any of the correlative measurements for clouds? SAGE II and III provide separate cloud product which can be used for cloud screening, and OSIRIS v5.0x is already screened for clouds.

Section 3.4: Can you provide referenced uncertainty estimate for MAESTRO? The authors stated that MAESTRO have issues that affect its measurements. If these issues are significant and impacting the quality of the data, then the data is not suitable for validation studies.

Section 3.5: Is OSIRIS aerosol extinction profiles V6.0 a released or research product? Why not use V5.0x, which is the officially released and validated OSIRIS product? As noted by the authors, V6.0 is noisier and saturate at low altitudes. The comparison with OSIRIS V6.0 can also be affected by the angstrom coefficient used. The authors should be careful using an angstrom coefficient derived using long wavelengths to calculate the aerosol extinction at shorter wavelengths. OSIRIS uses a fixed aerosol model in version 5.0x which can be used to convert the aerosol profiles at different wavelengths.

Section 5: To establish a baseline accuracy of the AerGOM aerosol profiles, the authors need to discuss the comparison with each instrument separately, and in more details, citing reported biases to explain the differences.

I don't understand why SAGE II and III comparison show different biases, since both instruments agree well with each other. The difference might be related to AerGOM retrieval accuracy in southern hemisphere hight latitudes measurements. Also, why SAGE III comparison only used above 15km.

The comparison between old and new GOMOS profiles don't really prove the authors claim that AerGOM is an improvement over IPF (figure 3). The authors need to better support thier argument, maybe showing time series comparison with SAGE II.

Section 6: This section should be shortened to include comparison with SAGE II, SAGE III and OSIRIS only. The authors already shown that POAM III sunset and MAESTRO measurements have larger biases than SAGE and OSIRIS.

Figure 8 and section 6 is difficult to follow because of incorrect legend.

Conclusion: The authors should rewrite this section without bullets. Also, the authors need to provide recommendations of the wavelengths range useful for scientific studies.

Minor comments: Table1: Change measurements method "limb" to "limb scattering".

The authors need to provide better text and caption describing each figure.

Figures 3,5,6,7, and 8 x-axes range should be changed to [-100, 100]

Figure 8: The legend box shows wrong zones, please fix it.

---

## Referee Comment (RC3) · Anonymous Referee #2 · 19 Apr 2016

Review of Manuscript AMT-2016-27

AerGOM, an improved algorithm for stratospheric aerosol retrieval
from GOMOS observations. Part 2: Intercomparisons

by Robert et al.

**General Comments**

This paper describes results of a new algorithm – AerGOM – used to retrieve aerosol extinction profiles from the GOMOS stellar occultation data. AerGOM extinction retrievals are compared to both the operational GOMOS retrievals from the IPF algorithm and to coincident measurements of extinction at multiple wavelengths obtained from five independent satellite sensors. The paper attempts to characterize the dependence of the observed differences on key occultation parameters that define the GOMOS data quality. This study is useful in that it explores potential sources of random and systematic errors in the GOMOS AerGOM retrievals, which will be of interest to scientists interested in using that data product. The paper is suitable for publication in AMT and I recommend it be accepted for publication after the authors address the issues raised below.

My primary criticism is that the results of the data comparisons are presented for the most part as a simple catalog of observed differences; and often no coherent explanation is provided for the observed systematic differences between the AerGOM retrievals and the other measurements. Also, it is hard not to conclude from the results presented that the operational IPF data generally agree as well and often better with the correlative measurements than the AerGOM results. This issue should be addressed directly in the conclusions.

**Specific Comments**

Page 2, lines 4-5 – there must be a standard reference to the current GOMOS operational processing algorithm (IPF v6.01). It should be used here.

Equation (1) – I believe the symbol $\sigma_{ref}$ should be $\beta_{ref}$ for consistency.

Sect 2.3 –If the cause of the anomalous retrievals is due to low signal-to-noise for dim and/or cold stars, why do these same events not fail for the IPF retrieval? Nothing in the discussion indicates that the AerGOM method lowers the signal-to-noise compared to the operational retrieval (since presumably the same Level 1 transmission data is used as input to both algorithms this could only result from changes in binning or smoothing). If the lower signal stars cause more problems with the AerGOM algorithm compared to IPF it is presumably less stable in some way that needs to be explained.

Figure 2 – In this and subsequent figures the following labels are used – "Sage 2/3" and "POAM 3". These should really use the Roman numerals (e.g., SAGE II) for consistency with the text.

Page 6, Line 20, last sentence beginning "It is also important…". I am not sure what this comment means or why it's relevant. Since only spatially coincident measurements are used for the comparisons (as they should be) it is not clear how the high-latitude sampling of SAGE III and POAM III should introduce any spatial bias to the comparisons. The authors should clarify this comment or remove it.

Figure 3 – The lighter curves in the right column (Absolute Extinction) corresponding to the other instruments are almost impossible to see. Either get rid of all the grid lines on the plots or change these curves to make them more visible (e.g., same thickness as AerGOM only different line type or symbols)

Page 11, line 20 – The last statement that "…the GOMOS data used is exactly the same." is not strictly true according to the description in section 2.2, which says that data from spectrometer B1 has been included in AerGOM. Presumably these channels are not used in the IPF algorithm. Have you quantified the effect of adding this data into the aerosol fitting algorithm?  These wavelengths (750-776 nm) correspond to the oxygen A-band, which is a strong absorption feature and thus dominates the background aerosol extinction. Have you quantified the effect of adding information from this channel, and can you show that it does not it cause any bias?

Section 6.2 – I find the discussion of SZA dependence confusing or incomplete. All the coincident measurements being compared to AerGOM in this study are made in sunlight. Four of the 5 instruments use solar occultation and thus measure at 90 degrees SZA by definition. The fifth – OSIRIS – uses scattered sunlight on the dayside of the orbit. So the use of GOMOS nightside occultations for the coincidences introduces a time offset, in both absolute UT and local time. This time difference is smallest for the lowest-SZA GOMOS measurements. The authors rightly point out that there is no known diurnal variation in aerosol extinction and attribute any observed dependence on SZA to a scattered light artifact in GOMOS. The GOMOS team should be able to characterize that artifact and know how it affects the aerosol retrieval. Does it make sense that this artifact impacts the wavelength-dependent difference profiles in the way observed?

Section 6.3 – It is not obvious to me that anything useful is added by including this section. There does not seem to be any definitive conclusion as to the effect (if any) of the obliquity of the GOMOS occultations on the data comparisons. This section is really just a long list of observations from the various plots that don't tie together in a coherent way. I would therefore suggest the authors consider removing this section.

**Technical corrections:**

Page 10, Line 26 – I think instead of 'throughs' you mean 'troughs'.

Page 14, line 16 – Correct 'cases case.'

Before resubmission the document should be scrubbed for editorial corrections. There are other grammatical and spelling errors that I did not bother to point out.

---

## Author Comment (AC1) · 5 Aug 2016

**Response to Interactive comments from Referee #1 on: "AerGOM, an improved algorithm for stratospheric aerosol retrieval from GOMOS observations. Part 2: Intercomparisons" by C. E. Robert**

We would like to express our gratitude to Mr. Dörner for the valuable comments made to improve the manuscript.

**Text legend:**
Referee comment
Author reply
*Author addition to the original manuscript*

**1. Technical corrections**

1.1 "Page 2 Line 22: There are two different kinds of dashes here."

This has been fixed. We used the en dashes, as recommended in the guidelines.

1.2 "Page 5 Figure 1: Depending on the PDF Viewer, the B side of the figure has a thin grey frame."

We did not notice this before, but it is true that if using Acrobat Reader DC for instance, thin lines appear around Figure 1b. We modified the way we created the figure and it seems like the thin line is gone now.

---

## Author Comment (AC2) · 5 Aug 2016

**Response to Interactive comments from Referee #3 on: "AerGOM, an improved algorithm for stratospheric aerosol retrieval from GOMOS observations. Part 2: Intercomparisons" by C. E. Robert**

We would like to thank referee #3 for the useful comments made regarding the manuscript.

**Text legend:**
Referee comment
Author reply
*Author addition to the original manuscript*

**1. Major comments**

1.1 The paper title clearly states that the analysis presented are for data Intercomparisons not validation. I suggest that the authors provide a proper validation and discussions of the observed biases and data quality rather than just present data Intercomparisons. To do so, they need to use officially released and validated correlative measurements, and expand on the discussion, mainly in section 5. Furthermore, its not clear to the user what is the AerGOM end product, is it aerosol extinction at selected wavelengths or fitting parameters or both.

We used intercomparison because, as explained in the text, it is difficult to find data with which one can validate this dataset, although it is not impossible. There is the plan to write a manuscript in the near-future within the scope of the Aerosol_CCI project (that uses the current AerGOM dataset for the stratosphere) that will at least partly focus on the validation of the AerGOM/stratospheric Aerosol_CCI Level 3 dataset using lidar observations and balloon-borne measurements. This will be done with colleagues who are well versed in these techniques and can perform proper validations.

The closest to what we can find to a properly validated dataset derived from satellite instruments are SAGE II, SAGE III and OSIRIS datasets. The SAGE instruments have been used extensively, and OSIRIS agreement with SAGE III measurements at 750 nm is excellent and provides a better coverage. Therefore, we improved Section 5 of the manuscript by putting more emphasis on the known accuracy and precision of these measurements and discussed the results of the AerGOM intercomparisons with more reference to the literature available. We also provide more tentative explanation of the difference observed between the various datasets and AerGOM, and come up with possible ways to improve the dataset in the future.

Concerning the AerGOM end product, see 2.2 below.

**2. Specific comments**

2.1 Abstract: Abstract should include summary of key findings.

The abstract has been improved to include a summary of the important findings.

2.2 Section 2: The authors need to describe the new data product format and retrieved aerosol wavelengths, if any.

As there is not yet an official dataset available, the format is not yet fixed, although it will probably be netCDF. To clarify what is retrieved, I added a paragraph to section 2.2:
*Given that the aerosol spectral law chosen for the AerGOM processing is of degree N, the AerGOM data product consists of extinction values at N + 1 wavelengths, but can be interpolated at other wavelengths using equations 2 and 3. The data used for the current work is based on a quadratic polynomial in inverse wavelength, with 350 nm, 550 nm, and 750 nm set as reference wavelengths.*

2.3 Section 2.3, p5: "These profiles are easy to identify and were discarded for the inter-comparisons of this paper", is it also valid for section 6?

Yes, no anomalous profiles have been included in the analysis of section 6. We also added more information on anomalous profiles in section 2.3 (at the request of another referee):
*The reason for the retrieval of such profiles by AerGOM was due to a combination of low signal-to-noise ratio (SNR) of the transmittance at shorter wavelengths for dim and cold stars, and an inadequate a priori of gaseous and aerosol species. The operational retrieval sidestepped this issue by using first a DOAS method to retrieve NO2 and NO3, removing their contribution from the measured signal before carrying out ozone and aerosol retrieval. This problem has now been fixed by using full climatologies of gas and aerosol species as a priori for the spectral inversion.*

2.4 Section 3: Did you screen any of the correlative measurements for clouds? SAGE II and III provide separate cloud product which can be used for cloud screening, and OSIRIS v5.0x is already screened for clouds.

No we did not screen the correlative measurements for clouds, since we have yet to find a reliable way to flag cloud observations in the AerGOM data, although we are actively trying to find a good detection algorithm.

2.5 Section 3.4: Can you provide referenced uncertainty estimate for MAESTRO? The authors stated that MAESTRO have issues that affect its measurements. If these issues are significant and impacting the quality of the data, then the data is not suitable for validation studies.

Sadly, no information on the uncertainty of the data were provided. But as this is an intercomparison instead of a proper validation, I still think that it's worth looking into comparisons between MAESTRO and AerGOM, especially given the small and extremely constant variability between both datasets correlative measurements below approx. 25 km (see Figure 3).

**2.6 Section 3.5: Is OSIRIS aerosol extinction profiles V6.0 a released or research product? Why not use V5.0x, which is the officially released and validated OSIRIS product? As noted by the authors, V6.0 is noisier and saturate at low altitudes. The comparison with OSIRIS V6.0 can also be affected by the angstrom coefficient used. The authors should be careful using an angstrom coefficient derived using long wavelengths to calculate the aerosol extinction at shorter wavelengths. OSIRIS uses a fixed aerosol model in version 5.0x which can be used to convert the aerosol profiles at different wavelengths.**

A first version of this draft used data v5 but the OSIRIS team gave us the opportunity to look at this new dataset. OSIRIS aerosol extinction v6 is actually not an officially released dataset as far as I know, but it has been the object of a peer-reviewed publication as referenced in section 3.5 (Rieger et al., 2014). The main reason to look at OSIRIS aerosol data v6 was that we were curious to look at the spectral dependence derived from measurements and that was added to the dataset in the form of an Angstrom exponent, which is more realistic than the fix aerosol model of version 5. However, we also agree with the referee that we must be careful due to the way the Angstrom exponent was derived, hence the last paragraph of section 3.5.

I any case, comparison of both datasets against AerGOM give mostly the same results at 750 nm (with the largest difference of 10% around 17 km, see figure below), so we added the following text in section 3.5:
*However, the comparison results presented in this paper can be generalized to OSIRIS aerosol extinction v5 dataset at 750 nm, as there are very little differences between both datasets when it comes to AerGOM comparisons.*

[Figure]

**2.7 Section 5: To establish a baseline accuracy of the AerGOM aerosol profiles, the authors need to discuss the comparison with each instrument separately, and in more details, citing reported biases to explain the differences.**

Section 5 has been improved in that respect. We tried to analyze the bias for each instrument separately and compared the results with values in the literature, with a

particular emphasis on the SAGE instruments as those have been used in several studies. More tentative explanations for the discrepancies observed are also provided, with possible ways to improve the AerGOM dataset in the future.

2.8 I don't understand why SAGE II and III comparison show different biases, since both instruments agree well with each other. The difference might be related to AerGOM retrieval accuracy in southern hemisphere high latitudes measurements. Also, why SAGE III comparison only used above 15km.

SAGE II and SAGE III comparisons with AerGOM are in good agreement when seen from the point of view of latitude of observations. Collocations with SAGE III were only found in the southern hemisphere (-60° to -30°), but collocations with SAGE II were found in all latitude bands considered. I recopied the figures below, and if you look only at the green curves of SAGE II and compare those with the SAGE III comparisons, you'll see that both comparisons agree pretty well up to approx. 30 km. AerGOM is to blame, since comparisons between SAGE II and SAGE III for different latitude bands show a very good agreement. So indeed, there is some bias introduced when retrieving stratospheric aerosol extinction in the mid-latitudes with AerGOM. The reason for this discrepancy is however unclear.

This aspect has been added to the discussion in section 5.

[Figure]

Concerning the use of SAGE III data above 15km: as mentioned in section 3.2, it is recommended to only exploit the SAGE III aerosol extinction 385 nm measurements above 16 km hence the cut-off at that altitude, but due to a bug in our plotting routine, the cut-off was applied to all wavelengths. This has been fixed now.

2.10 Section 6: This section should be shortened to include comparison with SAGE II, SAGE III and OSIRIS only. The authors already shown that POAM III sunset and MAESTRO measurements have larger biases than SAGE and OSIRIS.

I understand the point of view of the referee, but the fact that some datasets are very biased is not an important aspect for this part of the work. I think that the more important criteria is that the data has a constant bias, whatever it is, as we try to see if this bias changes according to some parameters. It has been shown that the relative difference variability for AerGOM and MAESTRO is the smallest of all datasets (~25% from 10 to 25km and beyond, depending on the wavelength), therefore I would argue that this stability of the comparisons is good for such a study.

POAM III, on the other hand, suffers from very large variability at shorter wavelengths (< 500 nm) except between 18 and 20km. At longer wavelengths, the variability is still large but in the range of what we see in the other comparisons between 13 and 20km. Therefore, due to this large variability (probably linked to the limited number of collocations), POAM III data is not taken into account for Section 6 of this paper as we cannot rely on the reproducibility of the bias between AerGOM and POAM III. We added some text to clarify that point in Section 6, 2nd paragraph:

*Due to the very large variability of the comparisons between POAM III and AerGOM observed in the last section, POAM III results are not included in this part of the work.*

2.11 Figure 8 and section 6 is difficult to follow because of incorrect legend.

This has been corrected.

2.12 Conclusion: The authors should rewrite this section without bullets. Also, the authors need to provide recommendations of the wavelengths range useful for scientific studies.

Done.

2.13 Minor comments: Table1: Change measurements method "limb" to "limb scattering".

Done

2.14 The authors need to provide better text and caption describing each figure.

We slightly improved the caption text for Figures.

2.15 Figures 3,5,6,7, and 8 x-axes range should be changed to [-100, 100]

Done.

2.16 Figure 8: The legend box shows wrong zones, please fix it.

Done.

---

## Author Comment (AC3) · 5 Aug 2016

**Response to Interactive comments from Referee #2 on: "AerGOM, an improved algorithm for stratospheric aerosol retrieval from GOMOS observations. Part 2: Intercomparisons" by C. E. Robert**

We would like to thank referee #2 for the useful comments made regarding the manuscript.

**Text legend:**
Referee comment
Author reply
*Author addition to the original manuscript*

**1. General comment**

1.1: My primary criticism is that the results of the data comparisons are presented for the most part as a simple catalog of observed differences; and often no coherent explanation is provided for the observed systematic differences between the AerGOM retrievals and the other measurements. Also, it is hard not to conclude from the results presented that the operational IPF data generally agree as well and often better with the correlative measurements than the AerGOM results. This issue should be addressed directly in the conclusions.

Both of these aspects have been addressed in this version of the manuscript. Section 5 now includes more references to accuracy and precision estimates from literature for various datasets, and provides tentative explanations for the differences seen between AerGOM and other datasets. Differences between AerGOM and IPF results are also discussed at much greater length in a section dedicated to the subject (section 5.2).

**2. Specific comments**

2.1 Page 2, lines 4-5 – there must be a standard reference to the current GOMOS operational processing algorithm (IPF v6.01). It should be used here.

I am not sure that there is a standard peer-reviewed publication that focuses solely on IPF6.01. The closest would be Kyrölä et al. (2010) but I think that the latest GOMOS ATBD would be a better reference, as this is really all about IPF6.01. I therefore added a reference to this technical report in the paper on Page 2, lines 4-5. Here is the new addition to the "References" section:
*Kyrölä, E., Blanot, L., Tamminen, J., Sofieva, V., Bertaux, J. L., Hauchecorne, A., Dalaudier, F., Fussen, D., Vanhellemont, F., Fanton d'Andon, O., and Barrot, G.: GOMOS Algorithm Theoretical Basis Document, Tech. Rep. GOM-FMI-TN-040, 2012.*

2.2 Equation (1) – I believe the symbol $\sigma_{ref}$ should be $\beta_{ref}$ for consistency.

I agree. This has been changed in the manuscript.

If the cause of the anomalous retrievals is due to low signal-to-noise for dim and/or cold stars, why do these same events not fail for the IPF retrieval? Nothing in the discussion indicates that the AerGOM method lowers the signal-to-noise compared to the operational retrieval (since presumably the same Level 1 transmission data is used as input to both algorithms this could only result from changes in binning or smoothing). If the lower signal stars cause more problems with the AerGOM algorithm compared to IPF it is presumably less stable in some way that needs to be explained.

One aspect of AerGOM that is interesting is that it retrieves all species simultaneously, but this is also a drawback here. In the GOMOS IPF, the retrieval of NO2 and NO3 is done using a DOAS method (as noted in the text page 3, line 21) improving the retrieval of these species in cases where the SNR is low. Since we do not use such an approach, sometimes the values retrieved for the NO2 and NO3 (and ozone and aerosols as well) can be non-physical and much larger (or lower) than one would expect (as seen in Figure 1B). But we recently found a way to improve the retrievals and get rid of the anomalous profiles by using the climatology as a starting point for the retrieval at all altitudes (and not just the top altitudes), a very simple modification that solves this problem and improves the stability of the AerGOM retrieval. It should be noted however that despite this improvement, these profiles have very large errors and especially at shorter, do not have a large information content at low altitude.

We modified section 2.3 to explain a bit more why there were anomalous profiles for AerGOM and not IPF, and how we corrected the problem:

*The reason for the retrieval of such profiles by AerGOM was due to a combination of low signal-to-noise ratio (SNR) of the transmittance at shorter wavelengths for dim and cold stars, and an inadequate a priori of gaseous and aerosol species. The operational retrieval sidestepped this issue by using first a DOAS method to retrieve NO2 and NO3, removing their contribution from the measured signal before carrying out ozone and aerosol retrieval.*

*This problem has now been fixed by using full climatologies of gas and aerosol species as a priori for the spectral inversion. However, this finding prompted the consideration that some of the retrievals might be affected by occultation parameters such as star properties and solar zenith angle (SZA) that could lead to straylight, and occultation obliquity which is an important factor in the imperfect correction of atmospheric scintillation (Sofieva et al., 2009). Therefore, another aspect of this inter-comparison involves studying the consistency of the agreement of AerGOM aerosol retrievals with those of other instruments under various occultation conditions. Section 6 presents the results of these comparisons.*

We also modified section 2.2 to mirror this new retrieval approach:

*Climatological profiles of various species are provided as a starting point for the non-linear Levenberg- Marquardt spectral inversion, leading to slant path integrated column densities and aerosol optical thicknesses at each tangent height.*

2.4 Figure 2 – In this and subsequent figures the following labels are used – "Sage 2/3" and "POAM 3". These should really use the Roman numerals (e.g., SAGE II) for consistency with the text.

This has been done. Note however that POAM III comparisons in Section 6 have been omitted, due to the large variability of the relative difference as seen in Figure 3 and also because POAM III data did not add much to the discussion.

2.5 Page 6, Line 20, last sentence beginning "It is also important...". I am not sure what this comment means or why it's relevant. Since only spatially coincident measurements are used for the comparisons (as they should be) it is not clear how the high-latitude sampling of SAGE III and POAM III should introduce any spatial bias to the comparisons. The authors should clarify this comment or remove it.

I agree that this sentence is confusing and not necessary. It has been removed.

2.6 Figure 3 – The lighter curves in the right column (Absolute Extinction) corresponding to the other instruments are almost impossible to see. Either get rid of all the grid lines on the plots or change these curves to make them more visible (e.g., same thickness as AerGOM only different line type or symbols)

Agreed. These curves now have a * symbol on top to help with the visibility, so I think that it should be much clearer to the reader. I didn't think a good idea to use dashed lines because it might have been confusing, as we already used dashed lines in the other panels to show IPF results.

2.7 Page 11, line 20 – The last statement that "...the GOMOS data used is exactly the same." is not strictly true according to the description in section 2.2, which says that data from spectrometer B1 has been included in AerGOM. Presumably these channels are not used in the IPF algorithm. Have you quantified the effect of adding this data into the aerosol fitting algorithm? These wavelengths (750-776 nm) correspond to the oxygen A-band, which is a strong absorption feature and thus dominates the background aerosol extinction. Have you quantified the effect of adding information from this channel, and can you show that it does not it cause any bias?

That is a good point. I removed that last sentence but added an entire subsection (5.2) dedicated to the discussion of IPF and AerGOM differences, as I think that this is an important point of the paper, which was also noted by the other referee.

Concerning spectrometer B1, it is indeed the case that IPF does not use data from this spectrometer. We actually avoid using pixels within the oxygen A-band to avoid the strong absorption features, so no bias should be introduced. I clarified this point by mentioning in section 2.2 the exact wavelength ranges that we use in spectrometer B1:
*The most important improvements implemented are: 1) the extension of the spectral range used for the retrieval using information from spectrometer B1 (**755-759, 770–775 nm**);*

2.8 Section 6.2 – I find the discussion of SZA dependence confusing or incomplete. All the coincident measurements being compared to AerGOM in this study are made in sunlight. Four of the 5 instruments use solar occultation and thus measure at 90 degrees SZA by definition. The fifth – OSIRIS – uses scattered sunlight on the dayside of the orbit. So the use of GOMOS nightside occultations for the coincidences introduces a time offset, in both absolute UT and local time. This time difference is smallest for the lowest-SZA GOMOS measurements. The authors rightly point out that there is no known diurnal variation in aerosol extinction and attribute any observed dependence on SZA to a scattered light artifact in GOMOS. The GOMOS team should be able to characterize that artifact and know how it affects the aerosol retrieval. Does it make sense that this artifact impacts the wavelength-dependent difference profiles in the way observed?

It is true that the discussion concerning the SZA was rather short. This section has been improved and extended to address the points you raise.

To summarize the additions to the section and answer your questions: it is the case that we assume that the dependence on the SZA is due solely to straylight. Actually, straylight should already be corrected in Level 1 GOMOS files, but there still seem to be some contamination in the data. The presence of straylight would increase the number of photons detected by the spectrometer, hence increasing the value of the transmission and decreasing the extinction. If this decrease in extinction is attributed to the aerosol (which has a slow varying spectral dependence unlike ozone, NO2 and NO3), then the comparison should show a decrease of the positive bias. If we assume the straylight to be more or less constant with altitude, its relative effect should be larger at high altitudes (> 25 km) and for longer wavelengths due to the generally much smaller aerosol extinction values typically found in such cases.

2.9 Section 6.3 – It is not obvious to me that anything useful is added by including this section. There does not seem to be any definitive conclusion as to the effect (if any) of the obliquity of the GOMOS occultations on the data comparisons. This section is really just a long list of observations from the various plots that don't tie together in a coherent way. I would therefore suggest the authors consider removing this section.

I only realized later that the methodology used in this section was flawed. Basically, one does not expect the obliquity to change the results of ensemble means, as was shown in Figure 7. The larger obliquity should introduce errors due to the inability to correct anisotropic scintillations, but these effects are random and observed at the level of one profile, not by averaging several profiles together (which would then just average the variability to nothing). Hence, this section should have shown instead a measure of the variability of the relative difference. But after looking at such results, there was even less to conclude from, and therefore I decided to follow the referee's recommendation and simply skip this section. We added some text in section 6 explaining this:
*A study of the effect of the obliquity on the bias was carried out, but the results did not bear any concluding evidence of a repercussion on the AerGOM measurements and were therefore omitted from the discussion.*

**3. TECHNICAL CORRECTIONS**

3.1 Page 10, Line 26 – I think instead of 'throughs' you mean 'troughs'.

Yes, this has been corrected.

3.2 Page 14, line 16 – Correct 'cases case.'

Done.

3.3 Before resubmission the document should be scrubbed for editorial corrections. There are other grammatical and spelling errors that I did not bother to point out.

We did our best to correct the paper, but without any specific comments, it is impossible to ensure that the paper is now error-free.